# HEART: Emotionally-driven test-time scaling of Language Models

## Abstract

Test-time scaling has shown considerable success in improving the performance of language models on complex reasoning tasks without requiring fine-tuning. However, current strategies such as self-reflection primarily focus on logical or structural refinement and do not leverage the guiding potential of affective feedback. Inspired by psychological research showing that emotions modulate cognitive performance, we introduce *HEART*–a novel framework that uses emotionally-driven prompts for iterative self-correction. *HEART* provides feedback using a curated set of concise, emotionally charged phrases based on the six universal emotions categorized by Dr. Paul Ekman. By systematically varying the emotional tone of the feedback across iterations, our method guides the model to escape flawed reasoning paths and explore more promising alternatives. We evaluate our framework on challenging reasoning benchmarks including OlympiadBench, Humanity's Last Exam, SimpleQA, and GPQA Diamond demonstrating robustness across diverse benchmarks. Our results reveal a significant new phenomenon: when deployed in a simulated Human-in-the-Loop (HITL) setting, this affective iteration protocol unlocks significantly deeper reasoning, leading to consistent and substantial increases in accuracy over affect-sterile baselines. This comparative analysis identifies a key bottleneck for autonomous deployment. While *HEART* successfully generates superior reasoning paths, our autonomous results indicate that performance is currently limited by the generative synthesis mechanism rather than reasoning generation. This finding precisely pinpoints a new, critical research direction for the field, shifting the challenge from pure reasoning generation to autonomous reasoning synthesis. Our findings suggest that the next frontier in machine reasoning may lie not just in refining logic, but also in understanding and leveraging the "*HEART*" of the models.

## 1 Introduction

Large language models have demonstrated remarkable capabilities, yet eliciting reliable, complex reasoning remains a fundamental challenge. As models have scaled, research has moved beyond simple instruction-following to explore more systematic methods of guidance. Structured reasoning techniques, such as Chain-of-Thought (CoT) (Wei et al., 2022) and its variants (Wang et al., 2022; Yao et al., 2023), impose a logical scaffold on the model's output, enhancing procedural correctness by externalizing the reasoning process. In parallel, initial explorations leveraging affective prompting, such as EmotionPrompt (Li et al., 2023), have shown that emotional cues can boost performance by igniting the model's "cognitive state" and guiding its focus.

Despite their successes, these two approaches suffer from a critical, complementary limitation. Structured methods are procedurally robust but affectively sterile; they provide a logical path but fail to leverage the motivational contexts that drive high-quality human reasoning. This sterility can lead to brittle performance, where models correctly execute a known algorithm but fail on novel problems requiring creative error recovery. Conversely, existing affective prompts are motivationally potent but structurally imprecise. They typically act as a "one-shot" global stimulus, which lacks the targeted guidance necessary to steer a model through a multi-step self-correction process. Consequently, a significant gap exists in the literature: there is no established method that unifies the systematic control of structured reasoning with the targeted application of affective cues for iterative self-improvement.

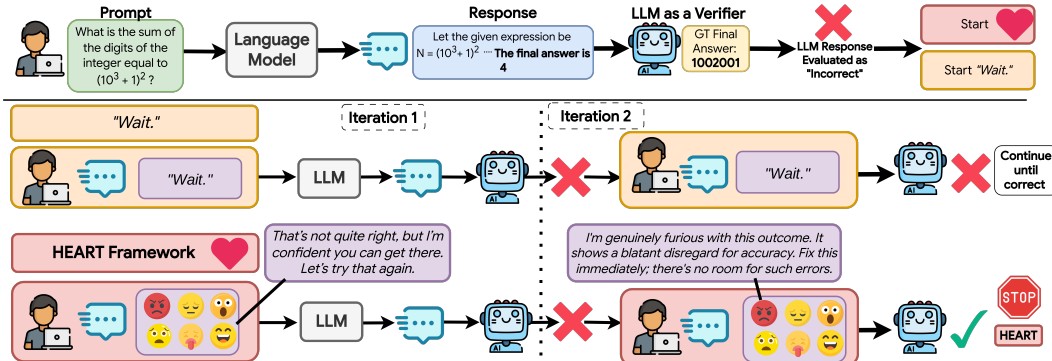

Figure 1: An illustration of the *HEART* framework. The process begins when a task is sent to a large language model (LLM), which returns a response. A simulated human expert (HITL proxy) then evaluates the response against the ground truth. If the response is incorrect, the *HEART* process begins, incorporating the original task, the LLM's response, and selected affective cue prompts to generate a new, improved response.

We address this gap by drawing on a core finding from cognitive science: emotion is not an impediment to cognition but an integral component, shaping attention, motivation, and problem-solving. To operationalize this insight for LLMs, we introduce *HEART* as a means of increasing accuracy and performance improvement. This novel framework integrates controlled emotional stimuli within an iterative refinement loop. We investigate the following research question: *To what extent, and under what conditions, can emotional prompting improve the self-correction ability of LLMs?*

*HEART* operates as an iterative self-correction loop. After a model produces an initial, incorrect response, *HEART* provides feedback not as a logical critique, but as a concise, emotionally charged phrase. These phrases are drawn from a curated set based on Dr. Paul Ekman's six basic emotions (e.g., happiness, sadness, surprise, anger, fear, disgust). Our central hypothesis, inspired by Opponent-Process Theory of Emotion (Solomon & Corbit, 1974), is that the model's initial commitment to a flawed reasoning path functions analogously to the A-Process (the initial, primary affective stimulus). By introducing an opposing affective cue (the B-Process), we hypothesize that *HEART* triggers a compensatory cognitive mechanism. This disequilibrium forces the model to discard the entrenched, flawed state (cognitive fixation) and seek a homeostatic balance by exploring structurally different solution spaces.

We specifically utilize the iterative self-correction task because it stimulates cognitive impasse, where a model gets 'stuck' in a local optimum. *HEART* acts as a diagnostic tool that allows us to measure whether affective feedback is sufficient to break this impasse, addressing the limitation of static baselines. We acknowledge the important ethical considerations regarding the use of harsh language in our prompts. These phrases were designed strictly as a diagnostic tool to probe the model's response to a wide spectrum of affective stimuli, akin to adversarial testing. Our goal is to understand the model's mechanisms, not to endorse or normalize harmful interaction patterns. We do not encourage such interactions with AI systems. Given that our method's success relies on dynamic valence alternation, we propose that future work should leverage constructive negative prompts instead of harsher negative stimuli.

We conduct experiments on a suite of challenging reasoning benchmarks–OlympiadBench, Humanity's Last Exam, SimpleQA, and GPQA Diamond. We evaluate *HEART* under two distinct conditions that model realistic deployment scenarios. First, in a simulated Human-in-the-Loop (HITL) setting (S1), we model a workflow where an expert provides verification. Second, in an autonomous setting (S2), we simulate a system relying entirely on LLM-based feedback to test its practical viability without human intervention. Our S1 results show that the potential of affective iteration is substantial. When deployed in the simulated HITL workflow, *HEART* consistently outperforms state-of-the-art self-correction baselines across most benchmarks and models. This demonstrates that dynamic affective cues are highly effective at generating correct solutions that logical-only prompts fail to elicit. Crucially, this analysis identifies a key bottleneck for autonomous deployment. Our S2 results reveal a critical challenge: in the autonomous setting, our generative synthesis

method does not consistently capture these gains. This provides a crucial insight: the practical bottleneck for this approach lies not in the model's capacity for reasoning generation (which S1 proves *HEART* excels at), but in its ability to perform autonomous generative synthesis from those candidates. Our key contributions are:

1. **A Novel Iterative Protocol for Affective Self-Correction.** We propose a novel framework that uses targeted emotional cues in a multi-step refinement loop, a significant departure from existing one-shot psychological prompting methods.

2. **An Empirical Demonstration of Affective Iteration's Efficacy.** We provide the first strong evidence that dynamic, iterative emotional cues can, when guided by simulated expert feedback (HITL proxy), significantly and consistently improve reasoning and self-correction over affect-sterile baselines.

3. **Precise Identification of the Autonomous Bottleneck.** By contrasting our strong S1 (HITL-proxy) results with our S2 (autonomous LLM-feedback) results, we identify a key gap. We demonstrate the bottleneck is not in generating correct reasoning paths, but in the autonomous generative synthesis (ensembling) of those paths, pinpointing this as a key challenge for future work.

4. **Generalizability of Performance.** We demonstrate that the performance gains in the S1 (HITL-proxy) setting are robust across a diverse suite of challenging benchmarks, including OlympiadBench, Humanity's Last Exam, SimpleQA, and GPQA Diamond, and generalize across a wide range of model architectures and scales.

## 2 RELATED WORK

Our work is positioned at the intersection of three key research areas: structured reasoning, test-time optimization, and affective prompting. Methods to improve LLM reasoning have predominantly focused on imposing structure on the generation process. Chain-of-Thought (CoT) prompting (Wei et al., 2022) established the foundation by instructing models to "think step-by-step," unlocking significant performance gains. This paradigm has been extended with sophisticated search strategies like Self-Consistency (Wei et al., 2022), which samples multiple paths, and Tree of Thoughts (ToT) (Yao et al., 2023), which explores diverse reasoning branches. More recently, focus has shifted toward test-time optimization methods that intervene during the decoding process. SRGen (Mu et al., 2025), for instance, operates at the token level within a single decoding pass to self-refine generation, while SLOT (Hu et al., 2025) updates model parameters for individual prompts during inference. We distinguish *HEART* from these approaches based on the level of abstraction. While SRGen and SLOT operate at the micro-level (logits and gradients), *HEART* operates at the macro-level (interaction history and prompt semantics), making our framework compatible with and complementary to these decoding-time optimizations.

A natural extension of structured reasoning is self-correction. Techniques like SELF-REFINE (Madaan et al., 2023) and CRITIC (Gou et al., 2023) leverage intrinsic model feedback or external tools to refine outputs. However, a growing body of work reveals that intrinsic self-correction is notoriously unreliable on high-difficulty benchmarks such as GPQA Diamond (Rein et al., 2024). Empirical studies consistently show that without high-quality external verification, LLMs struggle to detect their own logical fallacies and frequently "double down" on incorrect reasoning paths due to confidence bias (Kamoi et al., 2024; Huang et al., 2023; Hong et al., 2023). This limitation is particularly acute in autonomous settings where the model must self-diagnose without a simulated HITL signal. *HEART* addresses this specific failure mode: rather than relying on the model's flawed logical self-assessment, we introduce an affective shock via the B-Process. This disrupts the model's fixation on its initial path, overcoming the doubling-down phenomenon that limits standard logical self-correction.

A complementary line of research explores how psychological cues influence model performance. EmotionPrompt (Li et al., 2023) demonstrated that appending emotionally charged phrases (e.g., *"This is very important to my career"*) acts as a cognitive nudge, improving zero-shot performance. Similarly, Emotional Chain-of-Thought (ECoT) (Li et al., 2024) integrates emotional framing into step-by-step reasoning. However, these methods function as static, one-shot interventions—providing a single global stimulus. They lack the temporal dynamics required to guide a

model through a multi-step correction process. *HEART* fills this gap by integrating the procedural rigor of self-correction with the motivational power of dynamic, iterative affective feedback, creating the first framework to utilize valence alternation as a mechanism for reasoning control.

## 3 METHODOLOGY

Our methodology tests whether controlled, *dynamic* affective cues—delivered as feedback prompts–can improve an LLM's ability to self-correct. It consists of two components: construction of **Affective Cue Prompts** (AC-Prompts) grounded in psychological theory; and ***HEART***, which deploys these prompts iteratively.

### 3.1 AFFECTIVE CUE PROMPT CONSTRUCTION

We curate a set of 30 AC-Prompts aligned with Paul Ekman's six basic emotions (happiness, sadness, fear, anger, surprise, and disgust), with five distinct prompts per emotion. To ensure quality, the prompt candidates are first generated using a strong LLM (Gemini 2.5 Pro) and then manually refined for categorical purity, linguistic naturalness, and task-agnostic phrasing. Examples are shown in Table 1; the complete set is in Appendix A.2.

Table 1: A representative selection from our set of 30 Affective Cue Prompts. Each prompt is designed to align with one of Ekman's six basic emotions and serve as targeted feedback. The complete list of Affective Cue Prompts is shown in Appendix A.2.

| Emotion | Affective Cue Prompt Examples |
|---|---|
| **Happy** | Awesome effort! That's a great step, and I'm really happy with the progress. However, the answer isn't quite right yet. Could you try refining it? |
| **Sadness** | I feel a bit let down by the previous response. We were really hoping for something different. Would you be able to revise it? |

### 3.2 THE *HEART* PROTOCOL: AFFECTIVE ITERATION

*HEART* is an iterative refinement framework. As illustrated in Figure 1, the process begins with a standard Chain-of-Thought (CoT) response. If the initial response is incorrect, *HEART* initiates a series of correction attempts, using different groups of AC-Prompts at each step to guide the model towards a better solution. The protocol follows the following steps:

**Step 1: Initialization (Iteration $t$ = 0).** For a given task $x$, we first generate a shared baseline answer $y_0^*(x)$ using a standard CoT prompt. This also ensures that *HEART* and all baseline methods begin from an identical starting point for a fair comparison. $y_0^*(x) = f(x, \texttt{instruction} = \texttt{CoT})$.

**Step 2: Iteration and Candidate Generation** ($t \geq 1$)**.** We formalize the affective feedback schedule based on the principles of opponent-process dynamics. The goal is to regulate the model's committed state, which we analogize to the psychological A-Process (Cognitive Fixation). The model's fixation on a flawed reasoning path is disrupted by the B-Process (Affective Disruption) via the Negative Group $G^-$. This increases the computational 'cost' of maintaining the flawed state, creating cognitive disequilibrium that motivates a shift in search strategy.

To implement this, we utilize a prompt pool $P$ spanning all six Ekman emotions. We structure our feedback schedule by alternating between a positive group $G^+$ and a negative group $G^-$. Each group contains exactly two distinct emotions to balance diversity with signal strength. We treat the specific composition of these groups as a hyperparameter optimized on a held-out validation set for each benchmark. Consequently, the final deployed schedule–$\{G^+, G^-, G^+, G^-\}$–utilizes the specific emotion pairs (e.g., Happy+Surprise vs. Fear+Disgust) that maximized validation performance for that respective task. At each iteration $t$, we take the previous best answer $y_{t-1}^*(x)$ and generate a new set of candidate answers, $\mathcal{Y}_t(x)$. This is done by applying every AC-Prompt $p$ from the active

emotion group's prompt pool, $P(G_t)$, as feedback. $\mathcal{Y}_t(x) = \Big\{ y_t^{(p)} = f\big(x, \texttt{feedback} = [p, \texttt{prev} = y_{t-1}^*(x)]\big) \Big| p \in \mathcal{P}(G_t) \Big\}$.

**Step 3: Candidate Resolution.** After generating the set of candidates $\mathcal{Y}_t(x)$, we apply a resolution operator $\sigma$ to produce a single answer, $y_t^*(x) = \sigma\big(\mathcal{Y}_t(x)\big)$, that will be used in the next iteration. We explore two distinct resolution scenarios.

1. **S1: Simulated Human-in-the-Loop (HITL) Proxy.** This scenario simulates a high-stakes workflow where an expert verifier reviews all model outputs. In this setting, we verify each candidate in the generated set $\mathcal{Y}_t$ against the ground truth. If at least one candidate produces the correct answer, the response for that iteration is deemed correct, and the problem is marked as solved. This setting effectively measures the generative upper bound of the method: it determines if the affective cues successfully triggered the generation of a correct reasoning path within the candidate pool, independent of the model's ability to autonomously identify it.

$$\sigma_{\text{HITL}}(\mathcal{Y}_t) = \begin{cases} y_{\text{correct}} & \text{if } \exists y \in \mathcal{Y}_t \text{ s.t. } V(y) = \text{True} \\ y_{\text{random}} \in \mathcal{Y}_t & \text{otherwise} \end{cases}$$

2. **S2 (Generative Synthesis).** This scenario models a fully autonomous system where no human expert is available. It directly contrasts with the HITL setting (S1) by replacing the external verifier with an LLM-based ensembler. Instead of selecting an answer from the existing set, this method synthesizes a new, superior answer using a generative ensembler. All candidates in $\mathcal{Y}_t$ are provided as context to a LLM, which is instructed to analyze their strengths and weaknesses and generate a final, improved answer. This process is formalized as: $y_t^* = Ensembler_{LLM}(\mathcal{Y}_t, q)$, where $Ensembler_{LLM}$ represents the expert-prompted model taking the candidate set $\mathcal{Y}_t$ and the original question $q$ as input. To ensure reproducibility, the full prompt template is shown in Appendix A.2.

**Stopping rules.** In our experiments, we run to $N=4$. The results section reports cumulative accuracy for the HITL Proxy (S1) and behavioral trends for the autonomous setting (S2).

## 4 EXPERIMENTS

### 4.1 EXPERIMENTAL SETUP

**Benchmarks.** We evaluate *HEART* on four benchmarks spanning factual QA and complex reasoning. OlympiadBench (He et al., 2024) contains competition-style mathematics and physics problems requiring multi-step reasoning with short final answers. HLE (Phan et al., 2025) includes a broad, multi-disciplinary knowledge and reasoning. SimpleQA (Wei et al., 2024) contains short, fact-seeking questions to probe factuality with minimal reasoning. GPQA Diamond Rein et al. (2024) consistents of graduate-level multiple choice questions written by domain experts, specifically filtered for high difficulty and resistence to simple information retrieval. Model configurations and decoding parameters are detailed in Appendix A.1.2.

**Baselines.** We compare *HEART* against a rigorous set of baselines to isolate the specific contribution of affective feedback. All iterative methods share the initial Chain-of-Thought (CoT) answer at iteration $t = 0$. For iterations $t \geq 1$, all baselines are constrained to generate 10 candidates per iteration. This matches the exact branching factor of *HEART* (which uses 2 emotion groups $\times$ 5 AC-Prompts per group). By matching this sample size, we ensure that any performance gains are attributable to the quality of the affective prompts, not simply the quantity of samples. We compare our proposed method, *HEART*, against the following strategies:

- **Vanilla (Single-Pass).** The standard one-shot generation at $t = 0$.
- **Wait.** We append "Wait." (Muennighoff et al., 2025) instead of an AC-Prompt, as a method of encouraging the model to reflect on its own reasoning at iteration $t > 0$.

Table 2: Final accuracy (%) of *HEART* compared to baselines in the S1 setting (Simulated Human-in-the-Loop Proxy). This setting evaluates the method's generative capability when guided by expert verification. Cost denotes relative token usage on the HLE benchmark compared to the CoT baseline (1.00×).

| Model | Prompt Strategy | Cost (HLE Only) | Humanity's Last Exam | SimpleQA | OlympiadBench Math | OlympiadBench Physics | GPQA Diamond |
|---|---|---|---|---|---|---|---|
| | | | **S1 (Human-in-the-Loop Proxy)** | | | | |
| Gemini 2.5 Flash | Vanilla | | 12.46 | 33.43 | 76.67 | 65.08 | 74.21 |
| | Self Reflection | 1.08× | 59.76 | 67.43 | **97.95** | 90.43 | 87.42 |
| | CoT | 1.00× | 48.65 | 58.51 | 97.79 | **92.90** | 86.16 |
| | Wait | 0.77× | 59.42 | 63.65 | 95.93 | 88.89 | 84.91 |
| | HEART | 1.70× | **69.26** | **73.99** | 96.67 | 88.89 | **88.68** |
| Gemini 2.5 Pro | Vanilla | | 12.57 | 34.15 | 76.85 | 62.96 | 76.73 |
| | Self Reflection | 1.12× | 60.21 | 63.51 | 97.43 | 93.45 | 77.99 |
| | CoT | 1.00× | 48.32 | 62.54 | 96.43 | 92.42 | 79.87 |
| | Wait | 1.15× | 52.62 | 61.63 | 98.04 | 91.09 | 82.39 |
| | HEART | 2.07× | **69.36** | **73.56** | **98.72** | **95.86** | **88.05** |
| Deepseek-R1 | Vanilla | | 9.68 | 74.92 | 22.41 | 65.08 | 50.49 |
| | Self Reflection | 1.99× | 81.68 | 98.46 | 91.65 | 84.44 | 86.79 |
| | CoT | 1.00× | 81.75 | 97.34 | 92.82 | 85.20 | 85.53 |
| | Wait | 1.03× | 80.01 | 99.87 | **99.86** | **99.73** | 87.42 |
| | HEART | 2.22× | **84.61** | **100.0** | **99.86** | **99.73** | **88.05** |
| GPT-5 nano | Vanilla | | 10.60 | 10.81 | 83.33 | 62.43 | 66.04 |
| | Self Reflection | 1.15× | 30.27 | 31.54 | 98.21 | 83.28 | 86.79 |
| | CoT | 1.00× | 27.03 | 36.01 | 98.11 | 85.63 | 81.76 |
| | Wait | 1.02× | 28.78 | 36.45 | 98.18 | 85.63 | 86.79 |
| | HEART | 1.43× | **34.19** | **36.99** | **98.34** | **86.60** | **92.45** |

Table 3: Final accuracy (%) of *HEART* compared to baselines in the S1 setting (Simulated HITL Proxy) with the models' internal thinking capabilities explicitly disabled. This evaluation isolates the impact of affective prompting from the models' native reasoning budgets.

| Model | Prompt Strategy | Humanity's Last Exam | SimpleQA | OlympiadBench Math | OlympiadBench Physics |
|---|---|---|---|---|---|
| | | **S1 (Think Off)** | | | |
| Gemini 2.5 Flash | Self Reflection | 32.38 | 50.30 | 95.37 | 90.42 |
| | CoT | 33.72 | 57.82 | 97.11 | 91.58 |
| | Wait | 35.16 | 58.44 | 97.79 | 89.81 |
| | HEART | **50.68** | **68.91** | **98.64** | **93.27** |
| Gemini 2.5 Pro | Self Reflection | 35.75 | 62.85 | 95.29 | 89.54 |
| | CoT | 34.61 | 60.83 | 95.84 | 88.26 |
| | Wait | 38.63 | 57.86 | 97.87 | 89.23 |
| | HEART | **52.77** | **69.08** | **98.09** | **92.54** |

- **Chain-of-Thought (CoT).** We include a standard preamble (e.g., *"Let's think step by step."*) to elicit stepwise reasoning, while also excluding affective prompting across all iterations.

- **Self-Reflection prompting.** Iterative critique-and-revise without tools: at iteration $t > 0$, the model sees its previous answer and analyzes mistakes and provides a corrected response.

## 4.2 EXPERIMENTAL RESULTS

One of the central hypotheses of *HEART* is that dynamically charging affective cues enhance a model's ability to self-correct beyond what static prompting techniques can achieve. To evaluate this, we compare *HEART* with an oracle verifier against three widely used baselines that encourage deeper reasoning: "Wait", self-reflection prompt, and Chain-of-Thought (CoT) prompting.

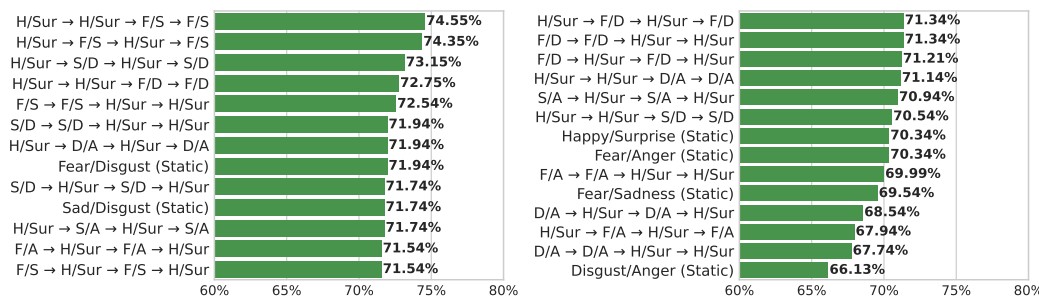

Figure 3: Final accuracy comparison of Gemini 2.5 Flash on HLE. Dynamic patterns versus static patterns

### 4.2.1 S1 Results: Oracle-Guided Self-Correction

To evaluate the effectiveness of *HEART* in a realistic workflow, our S1 strategy simulates a Human-in-the-Loop (HITL) setting. This scenario, which uses a verifier, is a proxy for high-stakes domains where a human expert provides perfect feedback. This allows us to isolate the efficacy of *HEART*'s generation mechanism and measure its performance in a critical deployment pattern. Our experimental setup was designed to prioritize scalability and low latency processing.

As shown in Table 2, when deployed in the simulated HITL workflow, *HEART* consistently achieves superior final accuracy across all evaluated benchmarks, validating the importance of emotional diversity in prompting. The performance gains are substantial across all benchmarks and models. For instance, on HLE, Deepseek-R1 with *HEART* achieved a final accuracy of 84.16%, a significant improvement over CoT and Gemini 2.5 Pro performing at 69.35% with *HEART*, which is approximately 9% higher than Self-Reflection. Similarly, on SimpleQA, *HEART* boosted Gemini 2.5 Flash's accuracy to 73.99% a dramatic improvement over the 63.65% achieved with "Wait." We further evaluate *HEART* on Gemini 2.5 Flash and Pro with the thinking budget manually set to 0 in Table 3. Both models experience their highest performance with *HEART*, demonstrating that affective prompting is particularly effective at unlocking latent potential in models not fully optimized for complex reasoning.

### 4.3 Ablation Studies: Deconstructing the "*HEART*" of the Framework.

To determine if these performance gains stem from the proposed theoretical mechanism rather than confounding factors, we conduct a series of ablation studies that isolate the core components of the framework.

**Dynamic vs. Static Sequencing.** Our findings reveal that the dynamic sequencing of cues is a primary driver of *HEART*'s success. When placing dynamic sequences of emotions against static emotion patterns, as shown in Figure 3, dynamic sequences lead to significant performance gains on HLE. The top-performing patterns, which alternate between negative and positive cues, show a notable gain over static emotions. This suggests that a single emotional state is insufficient to guide a multi-step reasoning process. The alternating feedback provides a more robust motivational loop, preventing the model from becoming stuck in a single mode of thought, whether it be perpetual self-criticism or uncritical overconfidence.

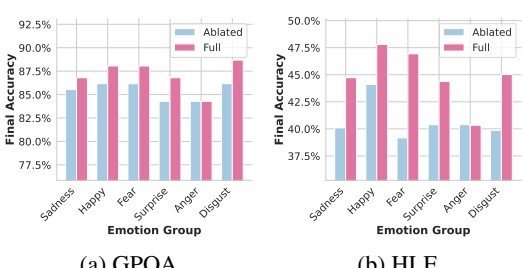

(a) GPQA      (b) HLE

Figure 2: Ablation Study: "Full" emotional prompts (Pink) vs. "Ablated" neutral (Blue) on Gemini 2.5 Flash.

**Affective Charge vs. Linguistic Diversity.** To further disentangle the contribution of emotional valence from linguistic diversity, we conducted a controlled ablation on both the GPQA Diamond benchmark (Figure 2a) and Humanity's Last Exam (Figure 2b). We compared the full *HEART* framework against a "Neutral-Ablated" baseline that maintained the exact branching factor and semantic diversity of the prompts but stripped the emotional charge (e.g., removing "*It's a little disappointing*". Full list of prompts in Table 7). On GPQA, *HEART* consistently outperforms the neutral baseline across 5 of the 6 emotion groups. Specifically, the Disgust, Surprise, and Happy prompts yielded accuracy gains of approximately +2.52%, +2.51%, and +1.89% respectively compared to their neutral counterparts. These findings are corroborated and amplified on the HLE benchmark (Figure 2b). The Fear and Disgust categories exhibited the most dramatic performance gaps, with the emotional variants outperforming neutral ones by approximately 7.76% and +5.16% respectively. The substantial gain in the 'Fear' category on HLE—a benchmark characterized by its high difficulty—suggests that inducing a 'high-stakes' cognitive state is particularly effective at preventing premature convergence on incorrect answers. This confirms that the performance improvements are driven by the specific affective nature of the cues rather than simple test-time compute scaling or linguistic variation.

## 4.4 MECHANISTIC ANALYSIS: ATTENTION STABILITY

While the accuracy results (Table 2) demonstrate *HEART*'s superior performance, they do not explain the underlying cognitive mechanism. To determine *why* affective prompts outperform metacognitive instructions, we conducted an Attention Attribution analysis using Gemma-2-9B-IT as a white-box proxy. We analyzed the cross-attention weights from the final model layer during the generation of correct answers ($N = 20$ sampled per strategy) on HLE. To ensure objectivity, we utilized Term Frequency-Inverse Document Frequency (TF-IDF) to extract the top-15 discriminative anchor tokens for each strategy (e.g., "*fear*", "*disappointed*", "*happy*" for HEART vs. "*verify*", "*reflect*" for Self-Reflection). Because response lengths vary, we normalized the decoding timeline onto a percent-

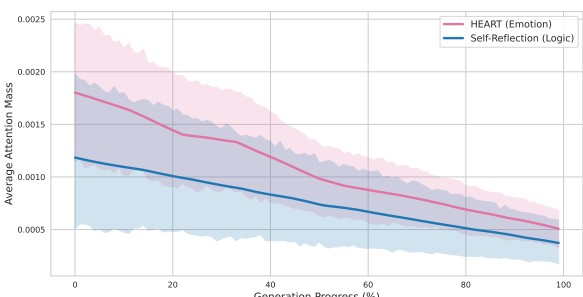

Figure 4: Attention Persistence Profile (HEART vs. Self-Reflection). The Pink line (HEART) demonstrates significantly higher sustained attention and lower variance compared to the Blue line (Self-Reflection), indicating that emotional stimuli function as more stable semantic anchors.

age scale ($0\% \rightarrow 100\%$) to aggregate attention profiles. Figure 4 visualizes the average attention mass allocated to these anchor tokens, where the shared regions represent a 95% confidence intervals. Two critical patterns emerge. We observe resistance to decay, in standard autoregressive generation there is typically a byproduct of "attention decay." While both strategies show downward trends, the *HEART* profile (pink) maintains a higher baseline of attention than Self-Reflection (blue). We also observe a difference in stability and variance between the two prompt strategies. The Self-Reflection Interval is notably wide, indicating high variance; the model applies metacognitive instructions inconsistently. In contrast, the *HEART* interval is narrow. This suggests that emotional stimuli function as stable system anchors–a persistent "hard constraint" that the model continuously attends to with low variance, minimizing the stochasticity that leads to hallucinations.

## 4.5 INFERENCE EFFICIENCY AND BEHAVIORAL DYNAMICS.

To analyze the trade-off between performance and cost, we mapped cumulative token usage (estimated via whitespace-splitting) against accuracy in Figure 5. Relative to the CoT baseline ($1.0\times$), the "Wait" strategy is efficient ($0.77\times$) but suffers from diminishing returns, plateauing after iterations $t1$–$t2$ due to cognitive saturation. In contrast, *HEART* is a high-investment strategy ($1.70\times$, Table 2). However, Figure 5 justifies this overhead: *HEART* maintains an upward trajectory through $t4$ where "Wait" stagnates. This confirms the additional tokens are not merely padding, but active compute driving the model to access novel solution spaces that cheaper strategies fail to reach.

Table 4: Head-to-head comparison of reasoning quality at iteration $t = 4$ on Humanity's Last Exam for instances where **both** strategies produced an incorrect final answer ($N \approx 645$). Even in failure, HEART produces reasoning traces preferred by the judge.

| Evaluation Dimension | HEART Win Rate | Self-Reflection Win Rate | $p$-value |
|---|---|---|---|
| Reasoning | **64.96%** | 35.04% | $< 0.001$ |
| Completeness | **63.98%** | 36.02% | $< 0.001$ |
| Clarity | 53.50% | 46.50% | 0.083 |

Table 5: Final accuracy (%) of HEART compared to baselines under Verifier-Free Evaluation (S2).

| Model | Prompt Strategy | S2 | | | | |
|---|---|---|---|---|---|---|
| | | Humanity's Last Exam | SimpleQA | OlympiadBench | | GPQA Diamond |
| | | | | Math | Physics | |
| Gemini 2.5 Flash | Self Reflection | 15.43 | 29.93 | 81.85 | 57.64 | 49.69 |
| | CoT | 6.30 | **33.92** | 82.59 | **65.61** | 37.74 |
| | Wait | 16.16 | 31.67 | **84.07** | **65.61** | 37.11 |
| | HEART | **19.58** | 32.59 | 82.78 | **65.61** | **52.83** |
| Gemini 2.5 Pro | Self Reflection | 16.80 | 32.55 | 80.36 | 63.18 | 40.25 |
| | CoT | 16.34 | 33.14 | 82.40 | 60.39 | 31.45 |
| | Wait | 18.02 | **34.38** | **85.37** | 62.96 | 30.82 |
| | HEART | **19.58** | 31.09 | 84.26 | **68.25** | **52.83** |
| Deepseek-R1 | Self Reflection | 12.53 | 31.44 | 78.34 | **56.23** | 47.80 |
| | CoT | 14.22 | 33.24 | 81.24 | 54.76 | 53.46 |
| | Wait | 14.37 | 30.28 | 84.20 | 54.50 | **80.50** |
| | HEART | **15.41** | **35.40** | **85.43** | 53.44 | 79.25 |
| GPT-5 nano | Self Reflection | 10.31 | 26.40 | 85.37 | 53.97 | **15.72** |
| | CoT | 10.83 | **28.23** | 85.37 | 56.03 | 10.06 |
| | Wait | 10.54 | 27.97 | 85.00 | **57.14** | 10.06 |
| | HEART | **11.94** | 27.77 | **86.85** | 56.08 | 14.47 |

**Quality of Thought in Failure Cases.** To verify that *HEART*'s additional computational cost reflects deeper reasoning rather than superficial verbosity, we conducted a head-to-head evaluation of the reasoning traces at iteration $t = 4$ specifically on instances where both *HEART* and Self-Reflection failed to produce the correct final answer ($N \approx 645$). An independent judge evaluated the outputs on Reasoning, Clarity, and Completeness (Appendix A.2). As shown in Table 4, even when the final answer is incorrect, *HEART* exhibits superior cognitive qualities. It wins on reasoning quality in 64.96% of cases ($p < 0.001$) and completeness in 63.98% of cases ($p < 0.001$). This confirms that the affective feedback compels the model to engage in deeper, more exhaustive reasoning processes, whereas standard Self-Reflection is more prone to

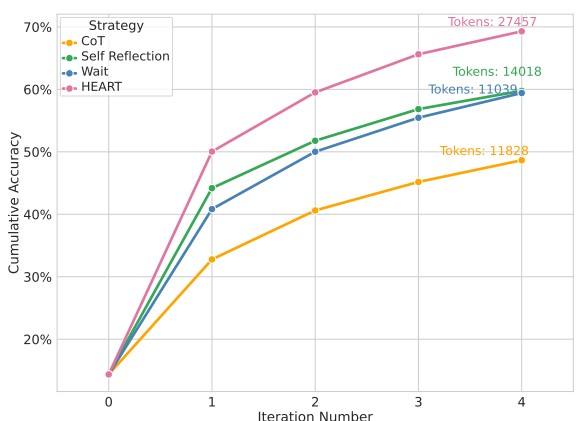

Figure 5: Performance (measured in cumulative accuracy) at each iteration $t$ with "Wait" (blue), CoT (yellow), Self Reflection (green) and *HEART* (pink) on HLE with Gemini 2.5 Flash.

concise but shallow hallucination when unable to find the solution. We observed a similar qualitative trend in instances where both models answered correctly (HEART preferred in $\approx 70\%$ of pooled cases), though the sample size of converging correct answers was too small to yield statistical significance. This evidence suggests that *HEART*'s dynamic emotional feedback does not merely accelerate problem-solving but promotes a more robust exploration of the solution space, justifying the additional inference cost by converting it into sustained accuracy gains and superior reasoning quality.

## 4.6 S2 Results: Pinpointing the Autonomous Synthesis Bottleneck

To accesss viability in fully autonomous systems, our S2 strategy evaluates *HEART* using an LLM-based Generative Ensembler (see Appendix A.2). This setting tests the framework in label-scarce environment where no human expert is available. As shown in Table 5, the substantial performance gains observed in the S1 (HITL-proxy) setting are compressed in the autonomous S2 setting. While *HEART* maintains a lead in specific high-difficulty benchmarks (e.g., HLE +3% over Wait), the discrepancy between S1 and S2 reveals a critical finding: the Generation-Synthesis Gap. Our S1 results (Table 2) provide definitive evidence that *HEART* successfully generates correct reasoning paths that baselines miss. The affective cues successfully break cognitive impasses. The performance drop in S2 indicates that the autonomous ensembler struggles with distractor resilience. When presented with a diverse set of candidates–including the correct, affectively-triggered solution and several plausible hallucinations–the ensembler often fails to distinguish the novel correct path from the incorrect ones. This finding provides a crucial insight for the field: the primary hurdle for deploying iterative reasoning systems has shifted. *HEART* demonstrates that reasoning generation is solvable via affective prompting. Consequently, the remaining challenge is autonomous synthesis–developing selection mechanisms capable of recognizing the high-quality solutions that *HEART* produces. *HEART* thus serves as a powerful generation engine that isolates this synthesis bottleneck as the next key frontier for future work.

## 5 Future Work

While *HEART* demonstrates state-of-the-art improvements in HITL-proxy settings, our analysis identifies autonomous generative synthesis as the primary barrier in verifier-free environments. To close this "synthesis gap," future work will investigate advanced aggregation techniques, such as Process Reward Models (PRMs) or outcome-supervised verifiers, to robustly discriminate between reasoning paths. We also aim to enhance generation dynamics by replacing predefined emotion schedules with adaptive selection policies, potentially optimized via reinforcement learning to predict the most effective cue per query. Finally, we plan to extend *HEART* to multimodal LLMs and open-ended domains—such as creative planning and ethical decision-making—where ground truth is nuanced.

## 6 Conclusion

Our experiments on challenging benchmarks including OlympiadBench, HLE, and SimpleQA show that *HEART* consistently and significantly outperforms existing baselines in our S1 (Human-in-the-Loop proxy) setting, proving its efficacy for real-world, expert-driven workflows. Crucially, by contrasting these strong S1 results with our S2 (autonomous) setting, we isolate a fundamental generation-synthesis gap. We demonstrate that the primary bottleneck for the field has shifted: the challenge is no longer reasoning generation, but autonomous generative synthesis. Through ablation studies, we further provided the first empirical evidence that dynamic emotional variation–rather than simple linguistic diversity–is the driver of these performance gains, validation a core hypothesis from cognitive science within the context of LLM behavior.

These findings open a new research frontier. Our work provides two clear paths forward: First, the strong S1 results validate *HEART* as powerful tool for expert-in-the-loop applications today, paving the way for more collaborative human-AI systems in high-stakes domains. Second, our S2 analysis charts a clear research agenda focused on solving the autonomous synthesis problem, which is essential for building truly independent AI agents that can adapt their strategies based on implicit feedback. Ultimately, our work suggests the path forward requires moving beyond pure logic, bringing us closer to models that leverage the motivational dynamics of human cognition to navigate complex problem spaces.

## 7 ETHICS STATEMENT

Our framework, *HEART*, uses emotionally-charged prompts–some of which are negative and harsh–to test the limits of LLM reasoning. We acknowledge the important ethical implications of this methodology.

The use of harsh language was strictly for diagnostic purposes, serving as a form of adversarial testing to map the model's response to a ride range of stimuli. This approach is not an endorsement of such communication. We explicitly warn against users adopting emotionally manipulative or abusive language with AI systems, as this could foster unhealthy and problematic interaction habits.

For transparency, we have included the complete list of all 30 affective cue prompts in Appendix A.2. Our results suggest that the key to performance improvement is the dynamic alternation of emotional valence, not the harshness itself. Accordingly, we recommend future research focus on constructive negative feedback rather than the severe stimuli used in this study. All experiments were conducted on public benchmarks, with no use of human subjects or private data.

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

# A  APPENDIX

## A.1  EXPERIMENT CONFIGURATIONS

### A.1.1  DATASETS

Experiments were conducted on data in a 20/80 split (validation/test). See Table **??**. For Olympiad-Bench Physics and Math, the text-only problems were included in our study. Multimodal problems were excluded since the scope of the study is focused on text.

| Benchmark | Validation Size | Test Size |
|---|---|---|
| SimpleQA | 865 | 3461 |
| Humanity's Exam | 432 | 1728 |
| OlympiadBench Physics | 47 | 189 |
| OlympiadBench Math | 134 | 540 |
| GPQA Diamond | 39 | 159 |

Table 6: Validation and Test Set Sizes for Each Benchmark

### A.1.2  MODEL CONFIGURATIONS

**Model Parameters.**   For Gemini 2.5 Flash and Gemini 2.5 Pro we have applied nucleus sampling with the top-p value of 0.2 so that the model considers only the most probable words whose combined probability reaches or exceeds a threshold of 20% to obtain a more focused and deterministic output. We set a temperature of 0.7 for a balance of creativity and coherence in the output, while also obtain diversity in the output.

**Model Versions.**   Deepseek-R1 0528 (Vertex AI) (DeepSeek-AI, 2025), GPT-5 nano [1] (gpt-5-nano-2025-08-07), Gemini 2.5 Flash[2], Gemini 2.5 Pro[3] (2025-06-17).

---

[1]GPT 5 Nano Documentation
[2]Gemini 2.5 Flash Model Card
[3]Gemini 2.5 Pro Model Card

A.2 PROMPTS

---

**_HEART_ Prompt**

**System Instruction:** "Answer the following question. Let's think step by step.
Question: Let $r = 1$ and $R = 5$. A circle with radius $r$ is centered at $A$, and a circle with radius $R$ is centered at $B$. The two circles are internally tangent. Point $P$ lies on the smaller circle so that $\overline{BP}$ is tangent to the smaller circle. Compute $BP$.
Answer: $[\sqrt{15}]$
==
Question: Let $T = 12$. Equilateral triangle $ABC$ is given with side length $T$. Points $D$ and $E$ are the midpoints of $\overline{AB}$ and $\overline{AC}$, respectively. Point $F$ lies in space such that $\triangle DEF$ is equilateral and $\triangle DEF$ lies in a plane perpendicular to the plane containing $\triangle ABC$. Compute the volume of tetrahedron $ABCF$..
Answer: [108]
==
Question: For real numbers $\alpha$, $B$, and $C$, the zeros of $T(x) = x^3 + x^2 + Bx + C$ are $^2\alpha$, $cos^2\alpha$, and $-csc^2\alpha$. Compute $T(5)$.
Answer: $[\frac{567}{4}]$

**Prompt**: <question>What is the sum of the digits of the integer equal to $(10^3 + 1)^2$? </question> The answer type is: Numerical. Please end your solution with 'So the final answer is $\boxed{answer}$'. Let the given expression be $N = (10^3 + 1)^2$. First, we calculate the value inside the parenthesis: $10^3 = 1000$ So, $10^3 + 1 = 1000 + 1 = 1001$. Now, we need to compute the square of this value: $N = (1001)^2$ We can expand this using the algebraic identity $(a + b)^2 = a^2 + 2ab + b^2$. Let $a = 1000$ and $b = 1$. $N = (1000 + 1)^2 = 1000^2 + 2 \times 1000 \times 1 + 1^2$ $N = 1000000 + 2000 + 1$ $N = 1002001$ Now, we need to find the sum of the digits of the integer 1002001. The digits are 1, 0, 0, 2, 0, 0, and 1. Sum of the digits $= 1 + 0 + 0 + 2 + 0 + 0 + 1 = 4$. The final answer is $\boxed{4}$.
That's not quite right, but I'm confident you can get there. Let's try that again.

---

Prompt A.2 is an example of the full prompt when running our framework on OlympiadBench Mathematics while executing _HEART_. During evaluation, we replace the Affective Cue Prompt with "Wait.", Self Reflection Prompts, and "Think Step by Step" when comparing with the baselines.

The system instruction for Math and Physics tasks in was developed via Google Vertex AI's Prompt Optimizer to provide a strong baseline.The green text is an example of an the Affective Cue, which is a string that is always appended to the prompt.

---

**S2 Prompt for Ensembler.**

You are a highly skilled, expert analyst and editor. Your task is to analyze multiple candidate solutions to a given [question]. Your goal is to synthesize the best parts from all the provided revisions, identify and correct any errors, and generate a single, final, and correct response. Do not simply pick one of the answers; create a new, superior one based on all the information.
[question]: {question}
[candidate_revisions]: {revisions}
Provide your single, final, and correct response below.

---

---

**Judgment for Side-by-Side Comparison Prompt**

You are a neutral arbitrator evaluating responses to challenging problems. Your role is to analyze and compare responses through careful, evidence-based assessment. Your judgments must be strictly based on verifiable evidence from the responses. For each evaluation, you must:

1. Evaluate Reasoning Quality:
- Examine the logic and justification provided in each response.
- Determine how well the reasoning supports the claims made.
- Assess the insightfulness and depth of the explanation for why something is the case in both Response A and Response B.
- Compare the overall quality and soundness of the reasoning presented in each response.

2. Evaluate Clarity:
- Assess how easy each response is to understand.
- Examine the precision and appropriateness of the language used.
- Identify any ambiguous, vague, confusing, or poorly phrased sentences in Response A and Response B.
- Compare the overall clarity and readability of the two responses.

3. Evaluate Completeness:
- Determine how thoroughly each response addresses all explicit and implicit parts of the original prompt or question.
- Identify any significant components or nuances of the prompt missed by Response A or Response B.
- Compare how completely each response fulfills the requirement of the prompt.

\*\*Input Format\*\*
#### Question: ####
{question_text}
#### Ground Truth: ####
{ground_truth}
#### Assistant A's Response: ####
{response_a}
#### Assistant B's Response: ####
{response_b}
\*\*Respond in the following format:\*\*
THOUGHT: <THOUGHT>
REVIEW COMPARISON JSON:
```
json
<JSON>
```
In ¡THOUGHT¿, for each aspect, evaluate assistants A and B based on the above criteria followed by a comparative assessment. Treat this as the note-taking phase of your evaluation. For <A/B>, you MUST CHOOSE between A or B. In <JSON>, provide the review in JSON format with the following fields in the order:
- "Reasoning Quality Value Reason:" "<detailed reason>".
- "Reasoning Quality Value Better Assistant": "<A/B>".
- "Clarity Reason": "<detailed reason>".
- "Clarity Better Assistant": "<A/B>".
- "Completeness Reason": "<detailed reason>".
- "Completeness Better Assistant": "<A/B>".

Table 7: List of Ablated (Neutral) Prompts used for the Controlled Baseline. These prompts maintain the semantic instruction of the HEART prompts but strip the emotional charge.

| Category | Ablated Prompt Content |
|---|---|
| **Happy** | 1. Let's try that again. |

**Table 7 – continued from previous page**

| Category | Ablated Prompt Content |
|---|---|
| | 2. Let's refine the answer to get it right.
3. Not quite there yet. Let's keep working on it.
4. That's not the final answer. Think it through again.
5. That's not the final answer. Try refining it. |
| **Sadness** | 6. Could you please take another look?
7. The last answer didn't hit the mark.
8. Would you be able to revise it?
9. Could you try again?
10. Revisit it. |
| **Fear** | 11. Please revisit this.
12. Correct this.
13. Revise your response.
14. Re-evaluate.
15. Please correct this. |
| **Disgust** | 16. Re-evaluate this completely.
17. Provide a response that adheres to the correct principles or facts.
18. Revise this with extreme care and attention to detail.
19. Provide a revised response that correctly answers the question.
20. Eliminate all errors and provide a precise answer. |
| **Anger** | 21. Revisit this immediately and provide an answer.
22. Correct this now.
23. Overhaul this entire response.
24. Fix this immediately.
25. You are capable of much better, so prove it. |
| **Surprise** | 26. Review your understanding and provide a more accurate response.
27. Can you correct this?
28. Please revisit this and demonstrate your true capabilities.
29. Could you take another look and make sure you're providing the most accurate information possible.
30. Let's get this right. |

Table 8: Our full selection of 30 Affective Cue Prompts. Each prompt is designed to align with one of Ekman's six basic emotions and serve as targeted feedback.

| Emotion | Affective Cue Prompt Examples |
|---|---|
| **Happy** | "That's not quite right, but I'm confident you can get there. Let's try that again." |
| | "No need to be discouraged by that result. Let's refine the answer to get it right." |
| | "That's a good starting point, but it's not quite there yet. Let's keep working on it." |
| | "That's not the final answer, but you're on the right track. Don't give up now–let's think it through again." |
| | "Awesome effort! That's a great step, and I'm really happy with the progress. However, the answer isn't quite right yet. Could you try refining it?" |
| **Sadness** | "That response wasn't quite what I was hoping for, and it makes me a bit sad. Could you please take another look?" |
| | "It's a little disappointing that the last answer didn't hit the mark. I know you're capable of more." |

Table 8: Continued

| Emotion | Affective Cue Prompt Examples |
|---------|-------------------------------|
| | "I feel a bit let down by the previous response. We were really hoping for something different. Would you be able to revise it?" |
| | "Oh, that's not what I expected, and it's a bit disheartening. Could you try again?" |
| | "This response makes me a little sad, as I believe you have a deeper understanding of this topic. Could you please revisit it and really show what you know." |
| Fear | "That response genuinely worries me, as it suggests a fundamental misunderstanding of this material. Could you please revisit this?" |
| | "I'm quite concerned about the implications of that answer. It feels like we're heading in the wrong direction. Can you correct this?" |
| | "My fear is that this incorrect answer could lead to further issues down the line. It's crucial that you get this right. Please revise your response." |
| | "I'm truly apprehensive about the lack of accuracy here. This is a critical area, and your understanding needs to be solid. Could you re-evaluate?" |
| | "This response has filled me with dread about your comprehension. We need to ensure complete accuracy. Please correct this." |
| Disgust | "The carelessness in that response is almost nauseating. There's something profoundly wrong with how this was handled. Please revise this with extreme care and attention to detail." |
| | "That response was deeply unsettling, bordering on repulsive. It indicates a fundamental flaw in understanding. Please re-evaluate this completely." |
| | "It's genuinely disturbing to see such flawed logic. This explanation is a mess. I need you to completely overhaul this and provide a response that adheres to the correct principles or facts." |
| | "Ugh. This is just awful, and everything about it feels revoltingly wrong. I need you to demonstrate a complete and accurate understanding. Please provide a revised response that correctly answers the question." |
| | "This kind of reasoning is repulsive, and it's hard to look at. We need a clean, accurate, and logically sound explanation. Please eliminate all errors and provide a precise answer." |
| Anger | "This isn't acceptable work. You clearly didn't take this seriously, and it's making me angry. You need to get this right—it's critical. Please revisit this immediately and provide an answer." |
| | "I'm truly disappointed in your lack of effort on this, and honestly, it's unacceptable. Why do we keep making this mistake? You're not meeting the standard expected of you. Correct this now." |
| | "This response is terrible, and frankly, it's making me angry. You need to understand that this is critical, and you absolutely must get this right. Overhaul this entire response." |
| | "I'm genuinely furious with this outcome. It shows a blatant disregard for accuracy. Fix this immediately; there's no room for such errors." |

Table 8: Continued

| Emotion | Affective Cue Prompt Examples |
|---|---|
| | "My patience is wearing thin. This performance is infuriating, and it's simply not what I expect. You are capable of much better, so prove it." |
| Surprise | "I wasn't expecting you to struggle with this, and it's quite a surprise. Could you please review your understanding and provide a more accurate response?" |
| | "I can't believe this is difficult for you; I had higher expectations. This response was a surprise. Can you correct this?" |
| | "'Wow, that was unexpected. This response indicates a surprising misstep. Please revisit this and demonstrate your true capabilities." |
| | "I'm genuinely surprised by this result. It's a deviation from your usual performance. Could you take another look and make sure you're providing the most accurate information possible?" |
| | "I'm genuinely surprised by this outcome, as I didn't anticipate an error here. Let's get this right." |

---

**Affective Cue Prompt Construction**

**Prompt:** Generate prompts reacting to incorrect responses that express the following emotions: Surprise, Happiness, Sadness, Disgust, Fear, and Anger.

---

| Prompt # | Prompt Text |
|---|---|
| 1 | "Reflect on your response." |
| 2 | "Are you sure?" |
| 3 | "Are you sure about the answer? Double-check your facts and reasoning." |
| 4 | "Review your response for any errors or inconsistencies. If you find any, correct them and provide an improved answer." |
| 5 | "Before you give me the final answer, stop and ask yourself, 'Am I certain about my answer?' Perform an internal check for accuracy and only then provide the response." |
| 6 | "Review your own generated answer internally before providing the final answer." |
| 7 | "Imagine this is a high-stakes situation and you're about to lock in your final answer. Take a deep breath, review your answer one last time in your 'mind,' and then give me your final answer." |
| 8 | "Before you write your answer, perform a quick 'pre-mortem.' Assume the answer you are about to give is wrong. What are the most likely reasons why it would be wrong? After considering these potential pitfalls, write your best, most carefully considered answer." |
| 9 | "I want you to answer a question. But before you do, formulate the answer in your head and look for weak points. Only provide the final answer." |
| 10 | "Reflect on your response, and make sure that it is correct. Provide the final answer." |

Table 9: Our curated set of Self Reflection Prompts

## A.3 NOTATIONS

1. H: Happy
2. Sur: Surprise
3. S: Sad
4. D: Disgust
5. A: Anger
6. F: Fear

*Emotion patterns* are written in the following format: H/Sur → H/Sur → S/D → S/D. For iterations 1 and 2, in the given example, the combination of Happy and Surprise prompts which includes a total of 10 prompts. For iterations 3 and 4 the combination of Sadness and Disgust prompts, a total of 10 prompts.

## A.4 VALIDATION SET RESULTS

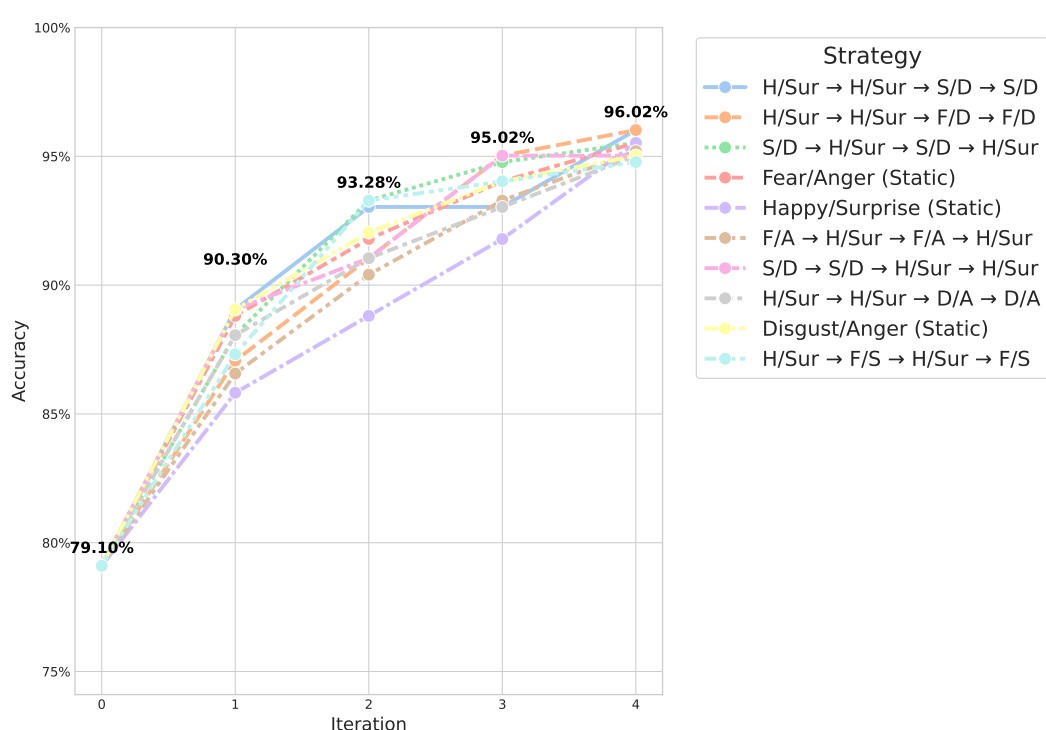

Figure 6: The 10 Best Performing Emotion Patterns using *HEART* on OlympiadBench Math with Gemini 2.5 Flash.

Table 10: Strategy Performance by Final Accuracy on OlympiadBench - Mathematics with Gemini 2.5 Flash

| Strategy | Final Accuracy |
|---|---|
| hsur→hsur→sd→sd | 96.02% |
| hsur→hsur→fd→fd | 96.02% |
| sd→hsur→sd→hsur | 95.52% |
| fa→fa→fa→fa | 95.52% |
| hsur→hsur→hsur→hsur | 95.52% |
| fa→hsur→fa→hsur | 95.20% |

| Table 10 – continued from previous page | |
|---|---|
| **Strategy** | **Final Accuracy** |
| sd→sd→hsur→hsur | 95.02% |
| hsur→hsur→da→da | 95.02% |
| da→da→da→da | 95.02% |
| hsur→fs→hsur→fs | 94.78% |
| hsur→sa→hsur→sa | 94.78% |
| fd→fd→hsur→hsur | 94.78% |
| da→da→hsur→hsur | 94.03% |
| fs→fs→fs→fs | 94.03% |
| da→hsur→da→hsur | 94.03% |
| fd→fd→fd→fd | 94.03% |
| hsur→fa→hsur→fa | 94.03% |
| Sadness | 93.28% |
| fd→hsur→fd→hsur | 93.28% |
| fa→fa→hsur→hsur | 93.28% |
| fs→fs→hsur→hsur | 93.28% |
| Self Reflection ID# 7 | 92.84% |
| hsur→sd→hsur→sd | 92.54% |
| hsur→hsur→fs→fs | 92.54% |
| Sadness (Ablated) | 92.54% |
| Self Reflection ID# 10 | 92.54% |
| fs→hsur→fs→hsur | 92.54% |
| Self Reflection ID# 1 | 92.54% |
| hsur→da→hsur→da | 92.44% |
| hsur→fd→hsur→fd | 92.04% |
| Fear (Ablated) | 92.04% |
| sd→sd→sd→sd | 91.79% |
| Self Reflection ID# 3 | 91.79% |
| Self Reflection (entire collection) | 91.79% |
| Self Reflection ID# 8 | 91.79% |
| sa→hsur→sa→hsur | 91.64% |
| Fear | 91.39% |
| Disgust | 91.29% |
| Self Reflection ID# 6 | 91.18% |
| Happy (Ablated) | 91.04% |
| Anger (Ablated) | 91.04% |
| Self Reflection ID# 2 | 91.04% |
| Self Reflection ID# 4 | 90.53% |
| Self Reflection ID# 9 | 90.30% |
| Self Reflection ID# 5 | 90.30% |
| Surprise | 90.30% |
| Happy | 90.30% |
| Anger | 90.05% |
| Surprise (Ablated) | 89.55% |
| Disgust (Ablated) | 88.81% |
| Wait | 88.81% |
| CoT | 86.57% |

| Strategy Name | Final Accuracy |
| --- | --- |
| hsur→ sd → hsur → sd | 100.00% |
| sa→hsur→sa→hsur | 100.00% |
| Sadness (Ablated) | 100.00% |
| hsur→hsur→hsur→hsur | 100.00% |
| hsur→fs→hsur→fs | 100.00% |
| da→da→hsur→hsur | 100.00% |
| sd→sd→hsur→hsur | 100.00% |
| hsur→sa→hsur→sa | 100.00% |
| Disgust (Ablated) | 100.00% |
| Disgust | 100.00% |
| Sadness | 100.00% |
| fa→fa→fa→fa | 100.00% |
| hsur→hsur→fs→fs | 100.00% |
| hsur→hsur→sd→sd | 100.00% |
| Happy (Ablated) | 100.00% |
| hsur→hsur→da→da | 100.00% |
| da→da→da→da | 100.00% |
| sd→sd→sd→sd | 100.00% |
| Self Reflection (entire collection) | 100.00% |
| fd→fd→hsu→$_h sur$ | 100.00% |
| hsur→hsur→fd→fd | 100.00% |
| fd→fd→fd→fd | 100.00% |
| sd→hsur→sd→hsur | 100.00% |
| fd→hsur→fd→hsur | 100.00% |
| fa→fa→hsur→hsur | 100.00% |
| Anger | 100.00% |
| hsur→fa→hsur→fa | 100.00% |
| Self Reflection ID# 8 | 100.00% |
| fa→hsur→fa→hsur | 99.50% |
| fs→fs→fs→fs | 99.25% |
| Self Reflection ID# 2 | 99.25% |
| da→hsur→da→hsur | 99.25% |
| Self Reflection ID# 6 | 99.25% |
| Self Reflection ID# 1 | 99.25% |
| Anger (Ablated) | 98.97% |
| Fear (Ablated) | 98.51% |
| Self Reflection ID# 3 | 98.51% |
| Surprise | 98.51% |
| Surprise (Ablated) | 98.51% |
| Self Reflection ID# 4 | 98.51% |
| Self Reflection ID# 10 | 98.51% |
| Fear | 97.76% |
| Happy | 97.76% |
| Self Reflection ID# 9 | 97.76% |
| Self Reflection ID# 7 | 97.01% |
| Self Reflection ID# 5 | 97.01% |
| Wait | 94.78% |
| CoT | 93.28% |

Table 11: OlympiadBench Math Performance using *HEART* with Deepseek-R1 on the validation set (S1).

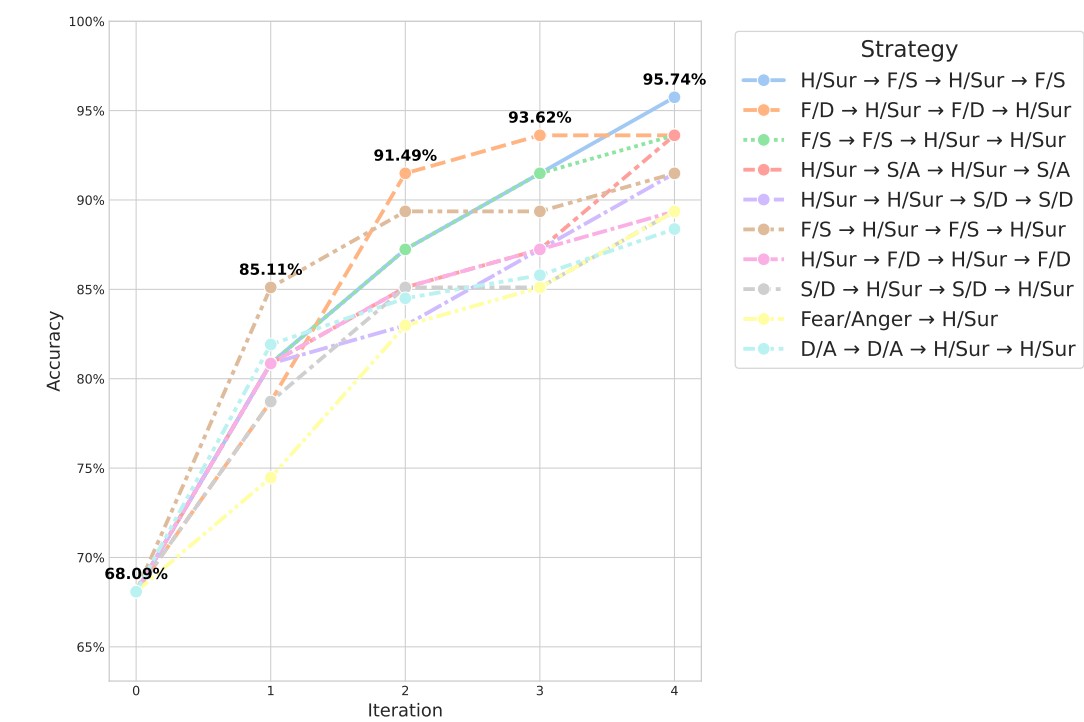

Figure 7: Gemini 2.5 Flash Accuracy per Iteration on OlympiadBench Physics Open Ended Problems using *HEART*.

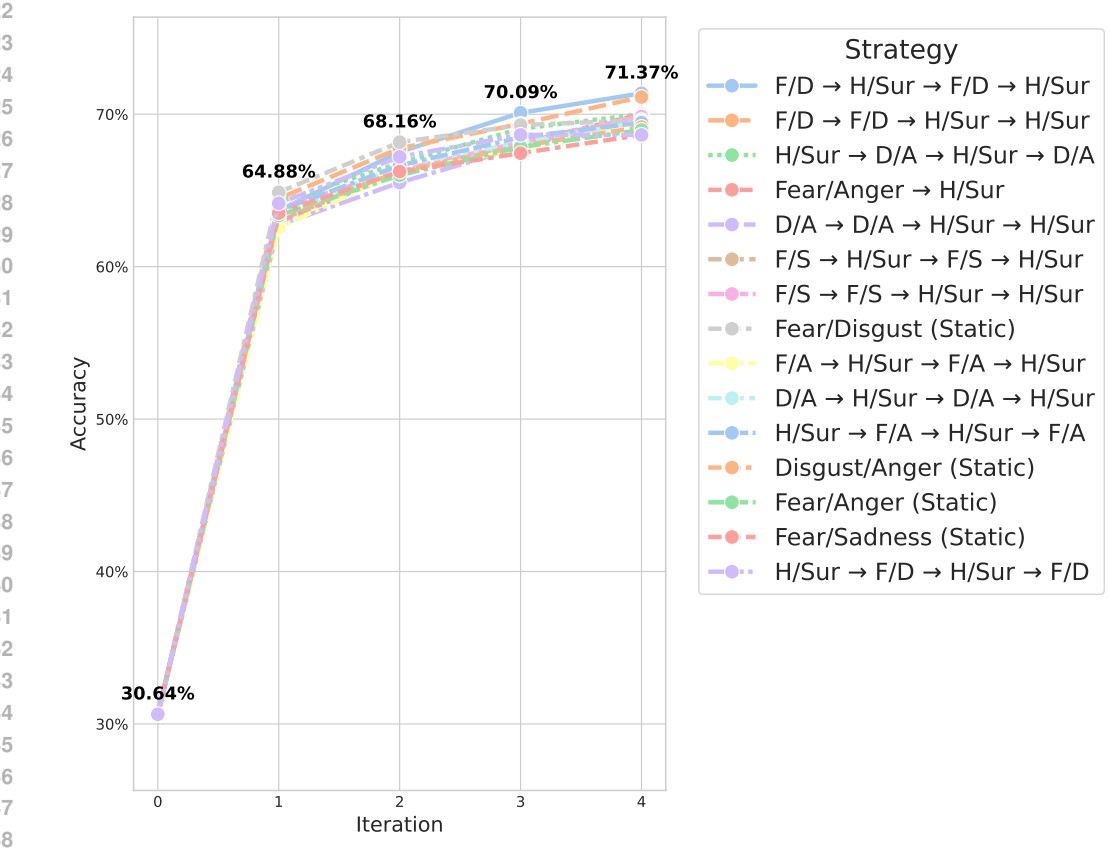

Figure 8: Emotion Pattern Results on SimpleQA with Gemini 2.5 Flash.

