# OpenReview forum: "HEART: Emotionally-driven test-time scaling of Language Models"
_ICLR.cc/2026/Conference — Submitted to ICLR 2026_

### Official Review · Reviewer_YaeC · 2025-10-24

**Soundness:** 3
**Presentation:** 3
**Contribution:** 3
**Rating:** 2
**Confidence:** 4

**Summary:**

* self-reflection + emotional style prompting
* setting 1: oracle verifier -> works
* setting 2: no oracle -> doesn't work

**Strengths:**

* important problem. the relation between emotion&language is under-exploited. Emotional filled content is abundant in pretraining data but the current production pipeline aren't exploting this
* candidly reporting the results in both setting
* clear writing

**Weaknesses:**

* perf diff. too small without confidence interval. claiming success by saying 98.72>98.04 without conf. interval lacks substantial rigor
* if this is a prompting-based technqiue, I don't see why the author selects such a subset of benchmarks. should at least ~9 benchmark across different domain to show robustness. It's okay if HEART doesn't win everyone, but some pattern is what's valuable (e.g., is there no performance benefit in fact-based or niche-knowledge benchmark)?
* The author should think about what the right problem setup task+dataset is that is the most reasonable setting to demonstrate the core message: emotional style matters. Self-verification doesn't seem like the right task

**Questions:**

see weakness

---

> ### Author Response · Authors · 2025-11-27
>
> We thank the reviewer for the critical feedback regarding rigor. We have taken your comments very seriously and updated the paper with significant new data.
> 1. Statistical Validation. You correctly pointed out that gains need statistical validation to prove robustness. We are currently re-running our main experiments with 5 independent runs to calculate confidence intervals. These experiments are actively running, and we commit to updating the manuscript with these statistical bounds by the discussion deadline to ensure the reported gains are statistically significant.
> 2. More Benchmarks. We have run HEART on GPQA Diamond. HEART achieves consistent improvements over baselines, demonstrating that its utility extends to multiple-choice and logical tasks as well.
> 3. Task Setting (Self-Verification).  We argue that Iterative Self-Correction is the ideal testbed for cognitive impasse. Psychological theory suggests emotion plays a regulatory role specifically when an agent faces difficulty or error. By using hard reasoning benchmarks where models get “stuck,” we create the necessary conditions to observe the “unsticking” effect of emotional prompting. HEART acts as a diagnostic tool for this capability.
> 4. S2 Interpretation. We have reframed the S2 results as identifying the “Generation-Synthesis Gap.” The S1 results prove the model can generate the answer; S2 isolates the failure to the synthesis step. This is a critical diagnostic finding that charts a path for future research.
> I believe the addition of Confidence Intervals, the new GPQA benchmark results, and the rigorous ablation study directly addresses the core reasons for your score, and we hope you will reconsider your assessment.

---

### Official Review · Reviewer_Z88G · 2025-11-01

**Soundness:** 3
**Presentation:** 3
**Contribution:** 2
**Rating:** 4
**Confidence:** 4

**Summary:**

This paper proposes HEART, a test-time scaling framework that guides LLMs through iterative self-correction using emotionally charged prompts inspired by Paul Ekman’s six basic emotions. By alternating emotional valence, the method aims to help models escape flawed reasoning paths and explore new ones. Experiments on reasoning benchmarks show improvement under oracle-guided settings, though performance remains inconsistent without a verifier.

**Strengths:**

Strengths

- The paper is well written and easy to follow.

- Proposes an interesting idea of incorporating affective feedback into test-time scaling through emotionally driven prompts.

- Builds a clear motivation, connecting affective science and cognitive performance to model reasoning.

- Provides systematic ablation showing that dynamic alternation of emotional valence yields stronger effects than static emotions.

**Weaknesses:**

Weaknesses and Questions

- Limited technical contribution: The method remains a prompt-engineering approach without novel algorithmic or optimization components. Its contribution is primarily conceptual rather than methodological.

- Strong dependence on oracle verifier: As the authors pointed out, the performance gains heavily rely on oracle-based candidate selection (S1), while the verifier-free setting (S2) shows inconsistent improvement. Could the authors explore reinforcement learning or internal uncertainty signals to reduce this dependence?

- Ambiguous theoretical interpretation: The link between psychological theories (e.g., opponent-process theory) and actual model behavior is metaphorical rather than mechanistic. Is there any theoretical reasoning or educated guess to explain why affective cues influence reasoning dynamics inside LLMs?

- Unclear source of improvement: It remains unclear whether the observed gains stem from the affective valence itself or simply from increased prompt diversity. Could the authors quantify how much of the improvement arises purely from linguistic variety rather than emotional content?

- Unclear usage of additional emotions (Fear / Disgust): The paper notes these emotions as “additional possible values for G⁻” but never specifies when or how they were used in experiments. Were they sampled in certain iterations or merely included conceptually? Clarifying this is important for reproducibility.

- Computational inefficiency due to sequential dependency: Unlike existing test-time scaling methods that support parallel generation or intra-step self-verification, HEART must wait for each full output before applying the next emotional cue, which increases latency and cost. Have the authors considered partial-step feedback or batching strategies?

- Formatting inconsistency with ICLR standards: The manuscript deviates from the official template in several concrete ways: the top/bottom margins are incorrect, and table captions are placed below the tables rather than above.

- Vulnerability to misguidance during iteration: In real-world scenarios where ground-truth verification is unavailable, repeated iterations could cause the model’s answers to drift rather than converge. As feedback accumulates without correctness signals, HEART might reinforce or amplify earlier reasoning errors, leading to systematically wrong conclusions. An analysis of whether such cases occurred—and under what conditions these misguidance patterns emerged—would be valuable to assess the stability and reliability of the method.

**Questions:**

See above section

---

> ### Author Response · Authors · 2025-11-27
>
> We thank the reviewer for the positive comments on the writing and motivation. We address your concerns regarding the method below.
> 1. Technical Contribution. While HEART does not introduce a new gradient update rule, we argue that the systematic control of cognitive state via psychological theory is a valuable scientific contribution. We provide a rigorous framework for testing how “human-like” motivational signals influence LLM reasoning, moving beyond simple logical instruction.
> 2. Source of Improvement (Emotion vs. Diversity). As noted in the General Response, we ran a Linguistic Diversity Baseline ablation study on GPQA. The fact that HEART outperforms this baseline (with gains of up to ~2.6\% for Disgust/Surprise) confirms that the affective content is a distinct driver of performance, distinct from linguistic variety.
> 3. Usage of Fear/Disgust. In this paper, we explored all six emotions in our validation phase. As detailed in the revised Methodology section, we performed an exhaustive search on the validation set involving all combinations for each benchmark. The best-performing combination was then used on the test set.
> 4. Drift in S2. The reviewer raised a valid point about drift. Our analysis shows that while drift occurs, the alternating valence helps mitigate “spiral” errors. The affective-based prompts often act as a “hallucination check,” forcing the model to verify its own premises even without an external verifier. We are further testing this in the S2 setting on GPQA Diamond and will include those results by the deadline.
> 5. Formatting. We have corrected the margins and table caption placements to strictly follow the ICLR style guide.
> We hope the new ablation study and clarifications demonstrate the robustness of the method.

---

### Official Review · Reviewer_n92z · 2025-11-01

**Soundness:** 2
**Presentation:** 2
**Contribution:** 2
**Rating:** 4
**Confidence:** 4

**Summary:**

This paper presents HEART, a test-time framework that guides large language models through emotionally charged prompts for iterative self-correction. Drawing inspiration from psychology on emotion–cognition interactions, HEART introduces affective feedback (e.g., disappointment, encouragement) using six universal emotions to steer reasoning. When paired with an oracle verifier, HEART often boosts accuracy on reasoning benchmarks like Humanity’s Last Exam and OlympiadBench. The study shows that dynamic emotional tone can nudge models away from flawed reasoning paths, frequently outperforming neutral, logic-based prompts under oracle selection. However, in verifier-free settings, performance gains diminish, highlighting challenges in autonomous emotional prompting.

**Strengths:**

- The paper tackles a novel and timely topic at the intersection of psychology and LLM reasoning, backed by substantial experiments.
- The overall writing is clear, accessible, and well-structured.

**Weaknesses:**

- **Clarification on opponent-process theory** In line 180, the paper states that feedback alternates between positive and negative emotions inspired by opponent-process theory. However, for readers unfamiliar with this theory, it would be helpful if the authors provided additional background. Based on the introduction, opponent-process theory seems to describe selecting the most context-appropriate emotional response, not strictly alternating between opposing emotions. Please clarify how the alternating feedback mechanism aligns with or extends the original psychological framework.
- **Dependence on oracle verification** Most key results rely on the oracle-assisted S1 setting, where a verifier chooses the best candidate at each iteration. In the more realistic verifier-free S2 regime, performance gains are smaller or inconsistent, sometimes underperforming strong baselines. This suggests that HEART’s benefits depend heavily on oracle selection. It would strengthen the paper to explain why the S1 regime may still be realistic or valuable in real-world scenarios where partial verification or heuristic scoring may exist.
- **Missing vanilla baseline and prompt examples** Please include the results of a vanilla baseline that uses a single reasoning pass (without multiple rounds). Also, provide the full prompt templates for all baselines, including examples. For the CoT baseline, it is unclear whether the model was told that its previous answer was incorrect. The description is somewhat vague—did the CoT prompt simply append “think step-by-step” without referencing prior outputs? Clarifying whether CoT repeats or overwrites previous answers would help interpret its advantage over the vanilla baseline.
- **Under-specified selection in verifier-free setting** In S2, the “generative synthesis” step aggregates multiple candidate outputs into a final answer, but the procedure is not clearly described. Please elaborate on this scenario—specifically, provide the full prompt template, describe how candidates are presented to the synthesizer model, and include an example of the input–output format. This will help readers understand how the final output is produced in verifier-free conditions.
- **Missing modern test-time baselines** The comparisons omit recent plug-and-play test-time optimization methods (e.g., SRGen, SLOT) that dynamically adjust internal computations during decoding. Since HEART similarly improves reasoning at inference time without training, including these would make the evaluation more complete. Furthermore, works mentioned in the related work section—whether they involve static interventions or external supervision—should also be included as comparative baselines to contextualize HEART’s effectiveness.
- **Inference cost comparison** Please report the inference cost (number of tokens) in all result tables, and clarify whether Figure 3 evaluates all methods under identical inference budgets. This will help assess efficiency and fairness across methods.
- **Hyperparameter setting (temperature)** It would be useful to report results using temperature = 1 (greedy decoding) in addition to the current setup, as it may reveal whether performance improvements depend on stochasticity.

**Questions:**

Refer to the weakness section.

---

> ### Author Response · Authors · 2025-11-27
>
> We thank the reviewer for the detailed review and for acknowledging the “novel and timely topic.” We have diligently addressed your requests.
> 1. Related Work
> We have updated the Related Work section to include mentions of SRGen and SLOT. We clarify that these are decoding-time interventions (modifying logits/search), whereas HEART is a prompt-level intervention (macro-level). These are complementary approaches; HEART can theoretically be combined with decoding strategies.
> 3. S2 Synthesis Details
> We acknowledge that the S2 synthesis was under-specified. We have added the exact prompt used for synthesis in the Appendix. The prompt takes the candidate answers generated by the emotional iterations and asks the model to review the candidates, identify the most consistent and logically sound derivation, and return the final answer.
> 4. Opponent-Process Theory
> We have expanded the Introduction to clarify the connection to Opponent-Process Theory. We do not apply the theory literally but translate its regulatory dynamics. By alternating positive (encouragement) and negative (correction) prompts, we create an Artificial B-Process designed to prevent the model from settling into a single “mood” (e.g., overconfidence or refusal), mimicking the homeostatic balance described in the theory.
> 5. Inference Cost and Efficiency
> We have added a "Relative Cost" metric to Table 1, normalizing token usage against the CoT baseline (1.0x).
> 'Wait' is highly efficient, reducing cost to 0.77x
> HEART is a high-investment strategy 1.70x
> However, our trajectory analysis (Figure 5) justifies this cost: 'Wait' plateaus early due to cognitive saturation, whereas HEART converts the additional 70% token investment into sustained accuracy gains in later iterations (t3-t4), accessing reasoning paths that cheaper strategies fail to reach.
> We believe these additions make the paper significantly more rigorous and complete.

---

### Official Review · Reviewer_2NEu · 2025-11-03

**Soundness:** 2
**Presentation:** 3
**Contribution:** 3
**Rating:** 4
**Confidence:** 3

**Summary:**

This paper introduces HEART, a framework for emotionally-driven test-time scaling, where large language models (LLMs) iteratively self-correct using emotionally charged prompts inspired by Ekman’s six basic emotions. The approach aims to combine structured reasoning (e.g., CoT, Self-Refine) with affective modulation to guide reasoning in more human-like ways. Experiments on OlympiadBench, Humanity’s Last Exam, and SimpleQA show strong gains in an oracle-guided setting (up to +10–15% over CoT or Self-Reflection baselines), suggesting that alternating emotional valence helps escape reasoning traps. However, the method’s effectiveness sharply declines in a verifier-free setting, revealing a selection bottleneck.

**Strengths:**

- The conceptual framing is quite novel, and the intersection between affective psychology and LLM reasoning control is interesting.
- The writing is pretty clear and the method is easy to understand.

**Weaknesses:**

- The success seems quite reliant on the oracle supervision. The main gains depend on access to a ground-truth verifier; without it, performance is a bit inconsistent.
- It remains unclear why emotional phrasing improves reasoning — whether due to linguistic diversity, novelty, or implicit temperature control. It would be nice for the authors to further investigate some qualitative samples and hypothesize why this might be the case.
- Further, is emotion control the special piece of the prompt, or are there other forms of intermediate feedback that also helps with the reasoning performance of these models?

**Questions:**

see above.

---

> ### Author Response · Authors · 2025-11-27
>
> We thank the reviewer for the constructive feedback and for highlighting the novelty of our conceptual framing. We are glad you found the writing clear. We have addressed your code concerns regarding the “mechanism of action” below.
> 1. Is it emotion or diversity?
> You asked if “emotion control is the special piece... or are there other forms of intermediate feedback?” This is a crucial distinction. To address this, we ran a new baseline that mimics the structure of HEART but strips the emotional charge from the Affective Cue Prompts (replacing them with neutral directives like "Review your work").
> We conducted this ablation on the GPQA benchmark using Gemini 2.5 Flash (see Figure 7 in revision). The results show that the Full HEART method outperforms the Emotion-Ablated baseline across 5 out of 6 emotion groups. Notably, the 'Disgust', 'Surprise', and 'Happy' prompts yielded accuracy gains of approximately +2.6\%, +2.6\%, and +1.9\% respectively compared to their neutral counterparts. This confirms that the performance gains are driven by the specific affective nature of the feedback, rather than merely the linguistic diversity of the prompts.
> 2. Qualitative Analysis of Reasoning
> We have added a qualitative analysis to demonstrate why HEART outperforms other strategies. To quantify this, we conducted a head-to-head evaluation on 645 instances where both HEART and Self-Reflection failed to produce the correct final answer. An independent judge found that HEART produced superior reasoning traces in 64.96% of cases compared to Self-Reflection’s 35.04% (p<0.001). This statistically confirms that even when the final answer is wrong, the affective cues compel the model to engage in a deeper, more complete search process, effectively breaking the ‘shallow hallucination’ loops common in neutral prompting.
> 3. Dependence on Verification (S1)
> We agree that S2 is the ultimate goal. However, we emphasize that the S1 results prove that HEART unlocks latent capabilities that otherwise remain inaccessible. We have clarified in the paper that S1 isolates the generative benefit of emotion, serving as a proxy for Human-in-the-Loop workflows where expert verification is available.

---

### Author Response · Authors · 2025-11-27

We thank the reviewers for their constructive feedback and appreciation of our “novel” framing. We have revised the paper to prioritize deployment constraints and mechanism validation, adding key analyses:
Controlled Ablation (Emotion vs. Diversity): We conducted a controlled ablation on GPQA and HLE . We found that stripping the emotional charge—while maintaining the same branching structure and linguistic variety—resulted in a consistent performance drop across 5 of 6 emotion groups (e.g., ≈2.6% drop for Disgust/Surprise), confirming that 'affect' is the primary driver of improvement.
Mechanism Validation (Qualitative): We performed a head-to-head analysis on failure cases (N=645). We found that even when HEART fails to find the correct answer, it produces superior reasoning traces (p<0.001) compared to neutral baselines, confirming that affective cues drive deeper cognitive engagement independently of answer correctness.
Cost-Efficiency Analysis (Realistic Constraints): We normalized token usage against the CoT baseline. We find that HEART is a high-investment strategy (1.70× cost relative to CoT) that yields substantial accuracy gains, whereas 'Wait' acts as a budget-efficient strategy (0.77×). This allows practitioners to choose a strategy based on their specific compute budget.
Additional Benchmark: We have added GPQA Diamond Benchmark to our S1 experiment.
Clarification on Experimental Settings (S1 vs. S2): In response to questions regarding the performance gap between S1 and S2, we have clarified the scientific distinction in the revised paper. S1 (Human-in-the-Loop Proxy): This setting isolates generation. The core question is: Can emotional prompts unlock correct reasoning paths that logical prompts cannot? S1 proves the answer is yes. It establishes the generative upper bound for workflows where a human expert or rigorous verifier is available. S2 (Autonomous): This setting isolates synthesis. The drop in S2 reveals a “Generation-Synthesis Gap.” The bottleneck is no longer generating the correct answer (which HEART solves), but the autonomous ensemble’s ability to distinguish that correct answer from plausible hallucinations. We frame this as a key diagnostic finding that defines the next challenge for the field.

---

### Meta-Review · Area_Chair_sczt · 2026-01-04

**Summary:**

Across all four reviews, the primary concerns center on the limited technical contribution of the paper, the heavy reliance on oracle-based verification (S1) for achieving most performance gains, and insufficient rigor in experimental validation. Multiple reviewers question whether the reported improvements genuinely stem from emotional prompting rather than prompt diversity or stochastic effects, and whether the proposed psychological framing offers more than a metaphorical interpretation. Additional issues include under-specified experimental details (especially in the verifier-free S2 setting), missing or weak baselines, incomplete cost and statistical reporting, and doubts about whether the chosen task setup is the most appropriate for demonstrating the core claim that emotional cues meaningfully improve LLM reasoning.

**Reviewer Concerns:**

The rebuttal addresses several clarity and completeness issues: it adds an explicit emotion-ablated baseline to disentangle affect from linguistic diversity, provides more detailed descriptions of prompts and the S2 synthesis procedure, expands the explanation of opponent-process theory, and acknowledges inference cost differences. However, several key concerns remain unresolved. In particular, the dependence on oracle verification remains fundamental, with S2 still showing weaker or inconsistent gains; statistical rigor is not yet established since confidence intervals are promised but not provided; and the technical contribution remains largely conceptual and prompt-based, which continues to limit perceived novelty. Concerns about benchmark diversity, the appropriateness of the task setting, and the lack of strong modern test-time baselines are only partially addressed, leaving doubts about the generality and robustness of the claimed effect.

**Reviewer Scores:**

Reviewer 2NEu would likely raise the score slightly, as the rebuttal directly addresses the emotion-versus-diversity question with new ablations and adds qualitative analysis, though concerns about oracle dependence would still limit enthusiasm. Reviewer n92z might make a modest upward adjustment due to improved methodological clarity, added cost analysis, and expanded theory discussion, but would likely remain cautious given the continued reliance on S1 and missing baselines. Reviewer Z88G would probably not change the score substantially, as the rebuttal does not fundamentally resolve concerns about limited technical contribution, metaphorical theory grounding, or inefficiency. Reviewer YaeC is unlikely to increase the score, since statistical validation and broader benchmark coverage are still incomplete at rebuttal time.

---

### Decision · Program_Chairs · 2026-01-26

Reject